# Spectral Techniques Applied to Evaluate Pavement Friction and Surface Texture

**Yuan Yan [1], Maoping Ran [1,2,\*], Ulf Sandberg [2], Xinglin Zhou [1] and Shenqing Xiao [3]**

[1]  School of Automotive and Traffic Engineering, Wuhan University of Science and Technology, Wuhan 430065, China; sahara1990@163.com (Y.Y.); zhouxinglin@wust.edu.cn (X.Z.)

[2]  Swedish National Road and Transport Research Institute (VTI), 581 95 Linköping, Sweden; ulf.sandberg@vti.se

[3]  School of Traffic science and Engineering, Harbin Institute of Technology, Harbin 150090, China; 19904638828@163.com

\*  Correspondence: ranmaoping@wust.edu.cn

**Abstract:** To study texture of pavement surfaces and its effect on pavement friction, this article features a field experiment conducted on in-service pavements to characterize surface texture by spectral analysis to substantiate links to friction values. Pavement friction was measured using a Mu-meter and British pendulum tester whereas texture data was collected using a stationary laser profilometer. Texture spectra were analyzed and expressed in third octave bands. The correlation between texture spectral levels and friction values at different speeds are discussed in the paper. Results show that the texture level, including spectral band levels, can well represent texture characteristics, as texture spectral levels have a good correlation with friction coefficient, especially the texture level of texture wavelengths at 1.25–12.5 mm versus $SFC_{sl}$ (representing the slope of the side force coefficient ($SFC$) versus speed), i.e., the slope of friction versus speed. This friction parameter gives better correlations with texture parameters than friction values at different speeds, which is believed to be an effect of the drainage caused by texture in that wavelength range.

**Keywords:** pavement; surface texture; spectral analysis; skid resistance; texture level

## 1. Introduction

Pavement friction is one of the key elements required for ensuring highway safety [1] as it represents the force that resists the relative motion between a vehicle tire and a pavement surface. The friction developed at the tire/pavement interaction is strongly affected by pavement surface texture.

Pavement surface texture is defined as the deviation of a pavement surface from a true planar surface within a specified texture wavelength range [2]. These deviations may be described in the frequency domain and are then defined by the texture wavelength, λ, and the peak-to-peak amplitude, A, of its components. This is referred to as the texture spectrum in a way similar to acoustic or vibration spectral analyses [3]. Pavement surface texture is generally divided into three categories [2]:

- Microtexture (λ < 0.5 mm, A is from 0.001 to 0.2 mm);
- Macrotexture (λ from 0.5 to 50 mm, A from 0.1 to 20 mm);
- Megatexture (λ from 50 to 500 mm, A from 0.1 to 50 mm).

Above the megatexture range, there is roughness, also known as unevenness, with λ > 500 mm.

The two ranges of texture that predominantly affect friction are microtexture and macrotexture [4]. Pavement texture is usually described by parameters reflecting the mean depth of the surface

macrotexture, such as mean texture depth (*MTD*), or mean profile depth (*MPD*) [5]. However, these parameters only disclose partial aspects of surface texture properties; e.g., *MTD* and *MPD* only reflect the volumetric, respectively the height property of the pavement surface [6] within a range covering parts of the macro- and megatexture.

Although amplitude parameters are indeed related to friction, however, thus far no consistent relationships have been demonstrated for pavement texture and friction, which depend on widely used traditional texture indicators, such as *MTD* and *MPD* [7], partly since friction is measured in many different ways. They are simply too crude as *MTD* and *MPD* are synthetic indexes of texture that cannot describe exhaustively the influence on the complex tire/pavement interaction. Two pavements exhibiting the same *MPD* can provide different friction values [8]. Crude texture amplitude values are not the only contributing features to friction, as shape, spacing, and distribution of the particles in the surface should also be considered [9]. Nevertheless, the *MPD* parameter has appeared to be very popular in current friction research and surveys, much due to it being easy to calculate based on surface profiles measured with mobile equipment. However, a more sophisticated way of studying the features of surface profiles on friction is by using spectral analysis of the surface profile. This provides a more detailed description of the texture; just like spectral analysis does in acoustics and other wave-related phenomena. As shown in the following, spectral analysis in fact played an important role when designing the *MPD* calculation procedure.

This study explores the relations between pavement friction and surface texture by using spectral analysis based on field experiments. Especially, the most important wavelength range for which texture has the best correlation with friction is indicated, as it is best suited to analyze and model surface texture influence on pavement friction.

Arguably, the first advanced attempts to relate friction with different pavement texture measures were made by Moore and presented in the 1960s [10]. However, these included simple measures such as the early versions of the sand patch method (MTD) and drainage-related measures such as the water outflow method, to name a few. Spectral technologies were not yet applied in friction analyses. The first attempt to relate friction with texture profile spectra was made by Henry and Hegmon in the 1970s [11], using a mechanical stylus profilometer. They presented spectra only visually for four pavements surfaces and correlated only overall rms values and their derivatives with friction.

A few years later, it was reported that there was a good correlation between the zero-intercept skid number ($SN_0$), i.e., the skid number extrapolated to zero speed, and microtexture based on the stylus profilometer, if the macrotexture range was omitted [12]. This confirmed the general view that microtexture is the influencing factor at low test speeds. Partly the same authors later published a review of candidate macrotexture and microtexture measurement methods for use at highway speeds with a recommendation of the more promising methods for further development, of which texture spectra was one, and supplemented this with a summary of the effects of pavement surface texture on skid resistance [13].

This was the research status at the time when this study was planned. Having access to the first high-resolution laser profilometers, a study was made in three countries, USA (inspired by the above mentioned references), Australia, and New Zealand, using the most common skid resistance testing equipment in each country and transporting the laser profilometers from Sweden and around the globe.

## 2. Spectral Analysis of Pavement Surface Texture

Frequency analysis (whether it be in terms of power spectra or power spectral density, PSD) can provide a detailed statistical description of the pavement texture profile. In 1976 one of the authors presented a method for spectral characterization of pavement surface profiles and presented several measurements based on a mechanical stylus profilometer [14], the same equipment that was later used to successfully correlate third-octave band spectra of tire/road noise with third-octave band spectra of texture profiles [15]. Another study had later in the mid-1980s indicated good correlation between pavement-induced fuel consumption with the related ranges of texture [16]. The latter was largely

facilitated by the access to a laser-based profilometer with performance superior to the earlier used stylus profilometers.

The technical specification ISO/TS 13473-4 [3] defines the methods that should be used to implement spectral analysis of pavement surfaces, based on measured two-dimensional surface profiles. The result of the frequency analysis is a texture spectrum, most conveniently expressed in one-third-octave bandwidth (or in another fractional-octave-band). However, this standard was not available at the time of testing; nevertheless, the measurements and calculations were made essentially according to the ISO technical specification.

## 3. Data Collection

### 3.1. Data Collection History

The data on which this study was based were collected already in 1988 by one of the authors (Sandberg) in a project involving similar measurements of pavement surface texture and friction in three international locations: Pennsylvania in USA, Victoria in Australia and in the South Island and North Island of New Zealand. However, the data analyzed here are only from one of these, namely New Zealand, essentially since these included the most varied textures and most measurements.

The project was initiated because portable and mobile (laser) texture profiling equipment had become available just a few years earlier, which made it feasible to study how texture influenced various parameters, such as pavement-related wet friction. Successful studies of texture spectral relations with tire/pavement noise and fuel consumption (i.e., rolling resistance) had already been done [15,16], and it was time to extend these studies to friction.

The results of these measurements were never published and are documented only in a project report to the sponsor [17]. However, the results were not lost, since they were presented and used in discussions and planning in international working groups, such as ISO, CEN(Comité Européen de Normalisation), and PIARC(Permanent International Association of Road Congresses), dealing with texture and friction; most importantly they inspired the large international project conducted by PIARC in the 1990s which resulted in (among other things) suggestions for an international friction index (*IFI*) and a macrotexture measure later standardized as the mean profile depth (*MPD*) [4].

Given the renewed interest in modeling of pavement friction; especially work currently active in China, it was decided to publish the results of the mentioned measurements even though it is now 30 years late.

### 3.2. Tested Pavements

The test objects included 21 different in-service road sites in New Zealand. The selection had been made in order to obtain as varied texture features as possible; such as maximum aggregate size, texture depth, varying proportions between micro-, macro, and megatexture, with a view to minimize intercorrelation (refer to Section 4.1). For example, mean texture depth (*MTD* measured with the "sand patch" method) varied between 0.57 and 4.34 mm, which is a very large variation.

Pavement types included dense asphalt concrete, porous asphalt concrete, cement concrete, interlocking cement blocks, surface dressings (also known as chip seals) and slurry seals. Ages ranged between new and 25 years old; all in service. This created a quite unique mix of very different textures, which is a prerequisite if one wants to distinguish between different spectral ranges (refer to Section 4.1).

### 3.3. Pavement Texture Measurements and Equipment

First, it is important to note that when these data were collected (1988), there were no standards available for texture measurements or data processing, except an ASTM(American Society for Testing and Materials; in recent years known as ASTM International) standard for the so-called sand-patch method (*MTD*). Equipment was largely analog; or sometimes partly digital.

Pavement texture was measured by a stationary (but easily portable) laser profilometer equipped with two laser sensors; one named "macro", covering texture wavelengths 2–500 mm, and one named "micro", covering 0.2–5 mm. The "macro" sensor was a Selcom 2005 (by LMI Technologies AB, Partille, Sweden) having a laser spot diameter of approximately 1 mm, while the "micro" sensor was a Remplir SD65-R12 (by Remplir AB, Stenkullen, Sweden) having a laser spot diameter of approximately 0.1 mm. The latter laser spot had greater diameter when it was outside of its focus area, which happened for part of the profile. In this project, therefore, the texture amplitude in the wavelength ranging below 0.5 mm was probably underestimated and of higher uncertainty (therefore, this range is cut in the diagrams below).

The sensors were carried over the measured profile on a "sled" on a circular steel beam driven by an electrical step motor via a chain; see Figure 1. A 1 m long profile ("trace") was recorded in each run. Sled speed was 50 mm/s when the macro sensor was used and 20 mm/s when the micro sensor was used. The motor had a tachogenerator that gave a trigger signal used to control the wavelength scale. The macro sensor had a vertical resolution of 8 µm while the micro sensor had a resolution of 1 µm. Together with the tape recorder, the noise floor (at standstill) was approximately at 1 µm (= 0 dB) in the 5 mm texture wavelength band for the micro sensor.

Calibrations were made by the use of two precision-made triangular waves cut in aluminum or bronze, one having a wavelength of 20 mm and 10 mm p–p amplitude, and the other a wavelength of 0.5 mm and a p–p amplitude of 0.25 mm (i.e., the slopes of the triangles were always at 45° to the vertical).

The performance of these laser sensors (including the tape recorder and the other processing system) was second to none of the profilometers used today, 30+ years later, with the exception of physical size, operating speed, and mobility.

For each site, 10 one-meter profile traces evenly spread over approximately 100 m in the wheel track where friction was measured were recorded on a 4-channel digital tape recorder (TEAC model R61, TEAC Corporation, Tokyo, Japan) allowing frequencies from 0 (DC) to 600 Hz; each profile with the length of 1 m. These were data recorded on-site, together with 250 mm long profile curves printed-out on tape.

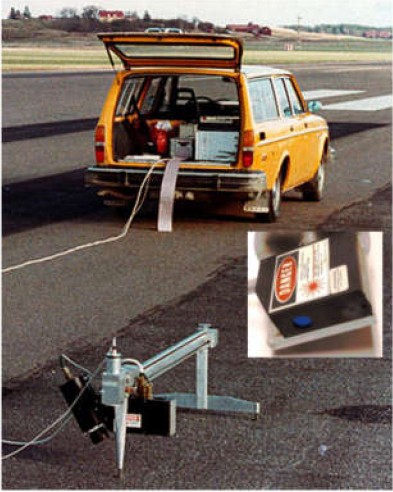

**Figure 1.** The VTI stationary profilometer, with its "macro" sensor mounted on the sled, and the "micro" sensor inserted in the photo, resized to approximately 5 times the scale of the rest of the photo. The photo was shot in another project.

Later, in the VTI laboratory in Sweden, after deleting invalid "spikes" (drop-out) values with an algorithm in a computer, these data were speeded up by a factor 10 and replayed in analog format into a third-octave band spectrum analyzer of type 2131 from Brüel and Kjaer (Brüel and Kjær Sound and Vibration Measurement A/S, Naerum, Denmark), which had been modified to allow one octave

lower frequencies (0.8–5000 Hz). Spectrum analyses were made by this device, according to IEC 60225:1966 [18].

The texture level of the corresponding wavelength bands λ was calculated in dB with the reference of 1 μm. The final texture level of each test site was the arithmetic average value (in dB) of all the 10 profile traces. See a typical spectrum in Figure 2.

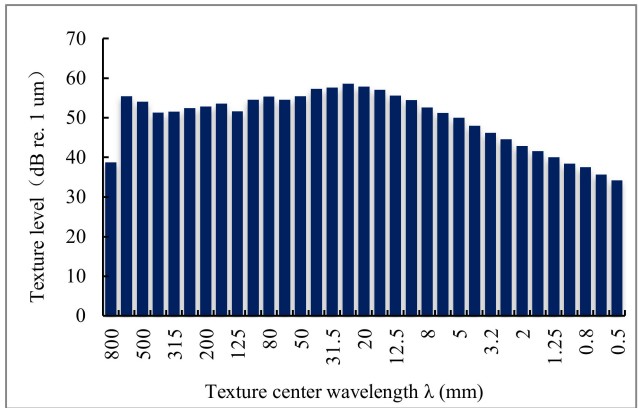

**Figure 2.** An example of a texture third-octave band spectrum (test site 1, a very rough chip seal). In this study, 95 % confidence limits were within ±1.5 dB in the range 2–100 mm and within ±2.5 dB at the longer and shorter wavelengths.

The limited evaluation length (length of one profile) was 1 m, which means that texture levels at wavelengths longer than 0.08 m had relatively high uncertainty, according to ISO/TS 13473-4. Despite this, when checking the standard deviations at the longer wavelengths, they suggested that the average spectral levels in the entire megatexture range were reasonably accurate.

Sand patch measurement (called sand circle in New Zealand) was also made on the same locations as the pendulum and profiling tests, i.e., at 10 locations per test site and the final value was the average of the 10 locations.

It should be mentioned that also the international roughness index (IRI) was measured. These data were not used here.

*3.4. Pavement Friction*

Pavement friction was measured by a Mu-meter. This is a small trailer towed by a car or van; see Figure 3. It had two measuring tires with 100 mm wide non-patterned (smooth) tread separated approximately 0.7 m with a toe-out of 15°. This means that the forces on these tires were side forces, which are measured by a load-cell. The Mu-meter is mostly used on airport runways, but nothing prevents it from working equally well on road pavements. The test speeds were 40, 60, and 80 km/h, and a minimum of five runs was made per each test location and speed. Water was sprayed on the surface from a nozzle in front of the tires, simulating a water depth of 1 mm. The output is a side force coefficient (*SFC*), which was recorded every 1 m, and the mean *SFC* over 100 m was calculated as the reported value for each test location. No temperature correction was made to *SFC*. It is generally assumed that the Mu-meter was sensitive to the macrotexture influence on friction, by a combination of drainage, which helped removing water from the tire/surface contact and by hysteresis losses. This is due to the relatively high speeds, relatively high water depth and the smooth non-patterned tires.

To complement these friction data, the British pendulum number (*BPN*), measured by the well-known British pendulum meter, was recorded on at least 10 locations per test site where at least five swings on each spot were made. The British pendulum meter is generally considered to supply a measure of pavement microtexture. This is due to the low speed at which the rubber slides over the surface, the relatively low amount of water applied and the low pressure by the rubber on the surface, which caused very little enveloping of the texture.

All the *BPN* values were corrected to a nominal road surface temperature of 20 °C utilizing the correction formula of reference [19] in Equation (1).

$$BPN = \frac{BPN_t}{1 - 0.00525(t - 20)} \tag{1}$$

The symbol t is the road surface temperature in °C.

## 4. Results

### 4.1. Intercorrelation in Texture

In order to be able to distinguish between the effects of the different texture wavelengths, the wavelength bands must not be too well correlated. Otherwise, if one detects a correlation between friction and one texture parameter, it is likely that the same correlation would be found with another texture parameter. Then it is impossible to tell which parameter is the best. One way to describe the texture internal relations is the intercorrelation. The intercorrelation is the correlation between the texture measured at one wavelength with the texture measured at any other wavelength.

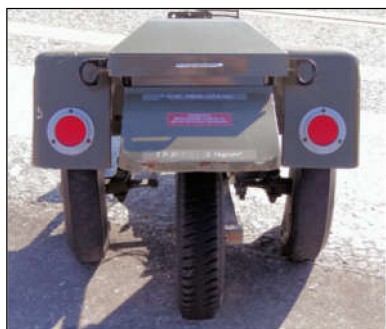

**Figure 3.** A Mu-meter with its two friction measuring tires at both sides of a free-rolling speed-measuring tire. The photo was shot in another project.

The intercorrelation found for the texture wavelength band levels in this study is presented in Figure 4. The intercorrelation graph was symmetrical around the diagonal, since the correlation coefficient between a and b is the same as between b and a. The color represents the value of the correlation coefficient, the darker the color, the higher the correlation coefficient.

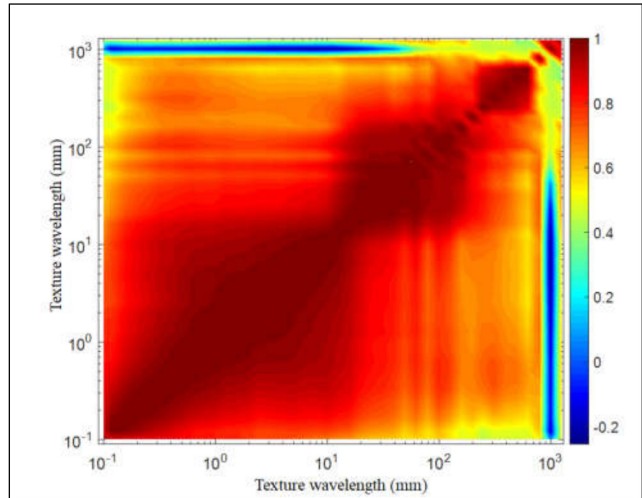

**Figure 4.** Intercorrelation between texture wavelength bands. The correlation coefficient values are illustrated by color according to the scale on the right.

One can see typical areas with high intercorrelation in Figure 4: there is one in the range 16–160 mm and another in the range 0.5–15 mm. This was probably due to the fact that a surface texture is generally dominated by its largest chippings. The largest chippings were normally around 8–16 mm on the selected test sites, which usually will give a fundamental wavelength of 10–20 mm. Subharmonics to this will mainly be in the range 16–160 mm and harmonics in the range 0.5–15 mm. So, in order to reduce intercorrelation between the test sites, it was necessary to vary the chipping size as much as possible.

The reason for the narrowed range around 200 mm might have to do with the inclusion of concrete paving blocks in the test, which of course have a pronounced periodicity. Ideally, one would like to have zero correlation in all other places in the tables except along the diagonal; but in practice, one may hope for a smaller dark area. Not only intercorrelation but also the range was an important feature for test site selection; especially, to include the finer part of macrotexture and touch on the upper microtexture range.

### 4.2. Analysis of Key Wavelength Ranges

Correlation analysis was applied to texture levels and friction values at the different test speeds, where *BPN* represented a rubber-to-surface speed of around 10 km/h (approximately 3 m/s), $SFC_{sl}$ represented the slope of *SFC* versus speed, which reflects how much friction is reduced by increasing speed. It was calculated by applying linear regression between friction values and speed. The results are shown in Figure 5, where the SFC slope, which was always negative, was inverted to fit into the same diagram as the rest of the curves.

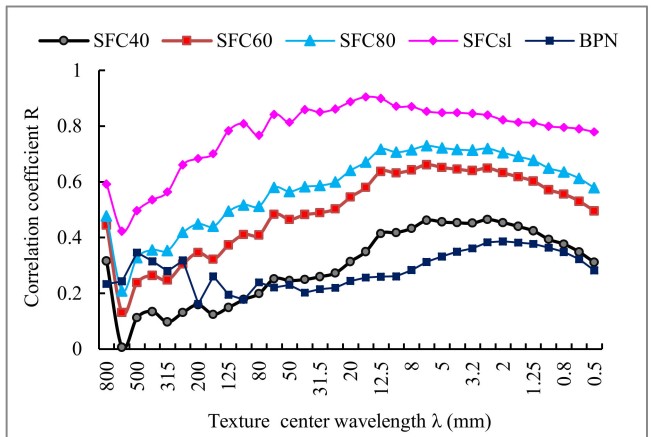

**Figure 5.** The correlation between friction values and road texture wavelength (*SFC*40 represents *SFC* at 40 km/h; *SFC*60 represents *SFC* at 60 km/h; and *SFC*80 represents *SFC* at 80 km/h).

Figure 5 shows that a positive correlation relationship existed between the two parameters representing friction and texture (when the *SFC* slope was inverted). From 0.5 to 600 mm of wavelength, the correlation coefficient R first increased to a peak value and then decreased, and this trend was repeated at all of the test conditions, although some small deviations occurred at the low speeds. The correlation coefficient R improved with the increase of speed. At each speed, wavelengths at which texture level and friction data with a correlation higher than 95 percent of the peak correlation coefficient R were selected as the "key texture wavelengths". At the speed of 80, 60, 40, and 10 km/h, the correlation coefficient R respectively reached 0.73, 0.66, 0.47, and 0.38 at the peak, which occurred at 6.3, 6.3, 2.5, and 2 mm of wavelength. Then the key texture wavelengths respectively were 2–12.5 mm, 2–12.5 mm, 2–6.3 mm, and 1.25–2.5 mm at each speed. Combining the key wavelength band at all the speed investigated, a recommend key wavelength band of 1.25–12.5 mm was determined. For $SFC_{sl}$, the speed gradient of friction, the correlation coefficient R reached the peak at 16 mm of wavelength (R = 0.90), and a wavelength range of 8–25 mm appeared to be the key wavelength range.

### 4.3. Correlation Between Various Texture Levels and Friction Data at the Test Speeds

In order to analyze if the key wavelength range can well represent pavement friction, texture levels of the key wavelength ranges were calculated with Equation (2).

$$L_{tx,i-j} = 10\lg\left( \sum_{m=i}^{j} 10^{\left(\frac{L_{tx,m}}{10}\right)} \right) \tag{2}$$

The texture levels of the key wavelength ranges were, respectively, indicated as $L_{\text{tx,1.25-12.5}}$ (including wavelength range 1.25–12.5 mm) and $L_{\text{tx,8-25}}$ (including wavelength range 8–25 mm). Other texture parameters are included to make the comparison of the correlations more complete. A couple of previous study results [8,20] have shown that the best correlation with pavement friction values were those containing the maximum energy content, so a range corresponding to that containing the maximum energy content, was identified as the range of wavelengths where the texture levels were within 3 dB below the maximum level. This range was designated as $L_{\text{tx,3dB}}$, also based on Equation (2). Besides this, megatexture level ($L_{\text{Me}}$, including wavelengths 63–500 mm) and macrotexture level ($L_{\text{Ma}}$, including wavelengths 0.63–50 mm) were also calculated based on (2), while the root mean square value of the profile (*RMS*) was calculated according to ISO/FDIS 13473-2. Finally, the set of texture parameters was supplemented by the mean texture depth (*MTD*).

Correlation analysis was applied individually to these six texture parameters and the friction values. The results are shown in Figure 6.

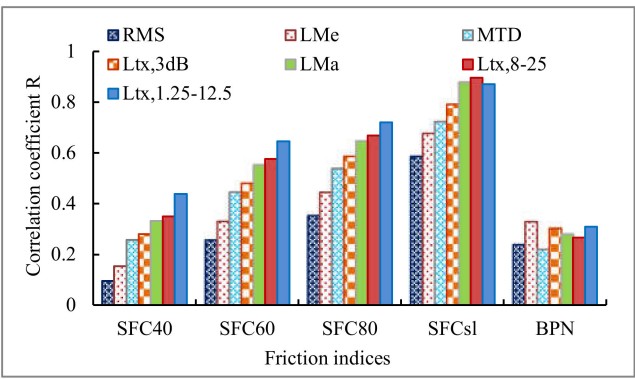

**Figure 6.** Pearson correlation coefficients R between six texture parameters and five friction indices.

As one can see from Figure 6, the results indicate consistent relations. The relationship between the newly determined texture descriptor $L_{\text{tx,1.25-12.5}}$ and the measured coefficient of friction appeared to be the best at the three speeds and is statistically significant. This indicates that this parameter could fairly well describe the relationship between texture and pavement friction.

The parameter $L_{\text{Ma}}$, which intrinsically reflects the same characters of pavement texture as the *RMS* (although on a logarithmic scale), got a substantially higher correlation coefficient R than *RMS*. This indicates the advantage of using logarithmic scales in such studies.

The friction index $SFC_{\text{sl}}$ shows very good correlation with most of the texture parameters, furthermore, the correlation coefficient R between this indicator and texture parameters was the highest (approximately 0.9) among all the friction indicators. This friction indicator was very useful as it represents the effect of speed on the friction values.

Finally, remember that the texture parameter, which is most popular currently, the *MPD*, did not exist at the time of these measurements, and lacking the original profile curves, it could not be calculated at this time and thus did not appear in Figure 6.

## 5. Discussion

It is important to note that these results were for friction measured by the Mu-meter, with its relatively small contact patch (less than $100 \times 100$ mm$^2$). A larger tire, such as the ASTM and PIARC reference tires in sizes typical of car or van tires might be sensitive to somewhat longer wavelength than these small ($16'' \times 4''$) Mu-meter tires.

Additionally, note that the intercorrelation in texture wavelength bands limits the possibility to distinguish between bands in their interaction with friction. In practice, it "widens" the curves in Figure 5 compared to the corresponding assumed "true" ones. An opposite effect may come from the higher uncertainty in the texture levels in the megatexture range (as mentioned above) and the same for wavelengths bordering the microtexture range.

As mentioned in the introduction, the project in which these data were collected (1988) inspired the PIARC project to repeat and enlarge this type of study to a much larger and international one that resulted in [4]. In this project, many more friction measuring devices were used, since the objective was to develop a common friction index to which results of any friction measuring device could be normalized.

In the PIARC project, with site selection made in a similar way, and with other types of friction testing equipment the intercorrelation in texture appeared basically similar to this project (Figure 4). For the friction slope vs. speed (average for all devices in the test), the correlation with texture spectra peaked with R = 0.88 at the 20 and 40 mm octave bands. This is quite similar to our results (Figure 4), except that the peak wavelengths were longer, which (as suggested above) might be due to the generally larger tires used in the PIARC study. It is logical that the hysteresis effect may be more sensitive to longer wavelengths when the tire/road contact patch gets longer. The key wavelength range was 2–80 mm; versus 8–25 mm compared to this study. This may reflect the wider range of test tires in the PIARC study.

The PIARC international experiment [4] suggested an international friction index (IFI) with the purpose to normalize results from different friction measuring equipment to an international index. The role of texture in this was introduced as a single-value variable ($T$) representing texture in the equation

$$S_p = a + b \times T \tag{3}$$

where $S_p$ is the speed coefficient of the friction coefficient (how friction is reduced by increasing speed), a and b are constants and $T$ is the texture measure (in practice an early version of the *MPD*). In [4] this worked rather well for normalizing the results of the testing devices. However, a follow-up project in 2001 named HERMES [21] suggested better results by a power model of the type

$$S_p = a \times (T)^b \tag{4}$$

When these options were tested in a separate project, with partly different devices, the results generally showed quite poor correlations and that the power model improved results only with test tires that had a smooth pattern (i.e., no pattern) [22].

It must be noted, however, that the speed coefficient of friction depends only indirect on macrotexture, as the primary dependence is on water drainage. If the test tire has enough water drainage capacity (such as the ASTM or PIARC tires with longitudinal grooves in the tread, or other tires with full-depth tread patterns), the drainage and thus macrotexture of the pavement surface is of only secondary importance since the test tire provides enough drainage. This is what was observed in [22]. The main effect of macrotexture for tires with full-depth tread grooves or other patterns were not on the speed exponent but more significantly on the overall speed-independent value, caused by the hysteresis effect (the latter of which is essentially speed-independent). These authors think that the IFI concept needs to be reevaluated to consider this.

For pattern-less (smooth) tires, the macrotexture effect is due both to the water drainage effect and the hysteresis effect and, therefore, the correlation between measured skid resistance values

and macrotexture can be as high as shown in this study. It follows, that the drainage properties of test tires are of crucial importance. As mentioned, the IFI concept needs to be reevaluated and one should find out a way to consider water drainage capacity in the tire together with water drainage capacity of the pavement texture.

In later years, attempts based on advanced and complex analysis of road surface profiles have been conducted to estimate indexes of macrotexture that are representative of the pavement friction values. Photometric techniques [23] and various transformation of profile curves, such as the Hilbert–Huang transform (HHT) have been used with varying success [24,25]. The latter is a way to distinguish between texture wavelength ranges, which is somewhat related to the spectral technology. However, in many studies, it is generally a problem with the limited databases used; generally too few and too different pavement surfaces, and/or the use of a special friction-measuring equipment, which is very different from when a real road vehicle with "regular" tires is braking.

When a tire rolls on the pavement surface not all the rubber tread gets in contact with the macrotexture represented by its profile. The tire envelops only the peaks of the texture profile and leaves a certain volume between the tread and the surface untouched. This enveloping effect has been studied in noise and rolling resistance research [26] and recently applied successfully also in friction research [27], where the enveloping procedure appeared to improve friction-texture correlation. In [28] a new parameter named 'texture drainage area' (TDA) was suggested, which represents the volume of air entrapped between the tire tread and the pavement surface. This can be further developed to include not only the air entrapped below the enveloping curve, but also the air in the tread pattern.

An excellent overview of the friction/texture mechanisms and including advanced modeling has been presented by Do and Cerezo [29]. They consider both micro- and macrotexture, with the former limited to the rougher wavelength range and the latter represented in three dimensions (3D). Interestingly, but logically, in laboratory experiments they have obtained high correlations between the sharpness (angles) of the asperities and friction parameters in order to represent the effect of the indenter principle. This is another way of representing the enveloping effect mentioned above.

The technology featured here, i.e., spectral representation of the texture profile, has not yet had an impact on international standards related to friction measurements, i.e., standards by ASTM or ISO. The *MPD*, which is closely related to the *MTD* and the $L_{\text{Ma}}$ used here, was not defined at the time of these measurements but was intended to replace these measures. In fact, the results of this study together with that of [4] were considered when the *MPD* measure [5] was developed in the early 1990s and revised in 2019, as the texture wavelength range on which *MPD* is based was selected to fit the best correlation of friction coefficients with texture wavelength bands, while at the same time fit the effective texture wavelength range of the *MTD* ("sand patch") measure. The *MPD* is now used in several ASTM and ISO measurement method standards and widely applied in pavement research, as well as in pavement surveys and management.

An interesting example is also the standardization of the IFI concept in ASTM E1960 [30] where MPD is an essential parameter. Recently, the IFI concept was revisited in an attempt in Europe to harmonize measurement results of European friction testing equipment, in the so-called ROSANNE project [31]. This was intended as a first draft for a European standard, which currently is under discussion. For the macrotexture influence they used the equation 4. The constants a and b were then assigned different values for different equipment (11 in total) as a result of the measurements and the harmonization came out in a way accepted by the project, albeit not as good as one would wish. For the side-force measuring devices, the standard deviation between the equipment was 0.035 (friction coefficient), while for longitudinal devices it was 0.038-0.051.

## 6. Conclusions

This work explored the use of spectrum analysis to investigate the texture–friction relationship. Texture spectra demonstrate how certain geometrical features in a surface can be described objectively.

The intercorrelation in the texture spectra appeared to be high in two wavelength ranges: 16–160 mm and 0.5–15 mm. When optimizing pavement performance, it is important to study this effect in order to see how sensitive various wavelength ranges or bands of macrotexture are in their influence on friction.

As shown in Figure 6, the correlations found in this study between the texture level of wavelengths within the range 1.25–12.5 mm and friction value are relatively high for 60 and 80 km/h and for the *SFC* slope, which means that when designing wearing courses of roads, texture should be especially concentrated to this range, if possible. This parameter is useful when developing models for pavement friction prediction. However, it is probable that the small tires used in this study resulted in a shift of key wavelengths towards lower wavelengths than would have been the case if larger test tires had been used. When searching for the optimum texture wavelengths to model texture influence on friction, especially the friction versus speed, one should keep this in mind.

The correlation between $SFC_{sl}$, the speed gradient of friction, and texture is relatively high, which means that it may be better to use this indicator in the analysis between texture and friction than directly applying the friction value at a certain test speed. This is in line with the conclusions of the great PIARC study made in the 1990s [4].

The increasing correlation between texture parameters and friction values with increasing speed is logical since macrotexture influences essentially the drainage properties of the pavement surface and these are increasingly important the higher the speed is. However, it is argued here that one must not forget the hysteresis effect on friction, i.e., the energy losses in the tire rubber due to deflections and indentations induced when a tire slips over a textured surface, which increases energy losses with increasing macrotexture.

The relationship between mega- and macrotexture and the *BPN* values is very poor and negligible. This is because the *BPN* is known to represent only the microtexture of the pavement surface, and microtexture could not be measured here except in the border region with macrotexture.

Finally, one must remember that this study used the Mu-meter—one of the most commonly used equipment for friction measurement—and that the relations between texture and friction here are valid mainly for this equipment. However, the PIARC study demonstrated that these relationships are in principle similar (but more or less pronounced) for all mobile measurement equipment.

**Author Contributions:** Conceptualization, U.S., and M.R.; methodology, U.S., M.R., and Y.Y.; validation, S.X., X.Z.hou and Y.Y.; formal analysis, M.R., and Y.Y.; investigation and all measurements, Ulf Sandberg; resources, M.R., and X.Z.; writing—original draft preparation, M.R., and Y.Y.; writing—review and editing, U.S., M.R., and Y.Y.; visualization, S.X., Y.Y.; supervision, X.Z.; project administration, U.S. and X.Z.; funding acquisition, Ulf Sandberg and X.Z. All authors have read and agreed to the published version of the manuscript.

**Acknowledgments:** The Chinese authors would like to acknowledge the financial support from the National Natural Science Foundation of China (NSFC; 51827812, 51778509, 51578430), the Hubei Provincial Natural Science Foundation of China (2018CFB293) and the overseas study program for young teachers of Hubei Province (201659194). Dr Sandberg acknowledges the support he received when he collected the texture spectra in project 87-01481 sponsored by Styrelsen for Teknisk Utveckling (STU); presently Vinnova (Sweden's innovation agency), Stockholm, Sweden. Some of the measurements were funded by the Road Research Unit of the National Roads Board of New Zealand, for which Dr Sandberg is also grateful.

**Conflicts of Interest:** The authors declare no conflict of interest.

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
