# Peer review of "Spectral Techniques Applied to Evaluate Pavement Friction and Surface Texture"

_coatings, doi:10.3390/coatings10040424_

Round 1

Reviewer 1 Report

The manuscript is based on Spectral Techniques Applied to Evaluate Pavement Friction and Surface Texture. Manuscript contains applicable resultst which have potential to bring an advancement into the studied subject. Discussion part is "making sense". However, there are present some points which decrease the overall quality of the manuscript and have to be corrected.

  1. The introduction part is too superficial. Introduction does not provide sufficient background and includes only few relevant references. Thus, introduction has to be given in a larger extent and containing additional relevant references.
  2. Lines 108, 109, 130 and 135: For each used instrument, the supplier, city and country of the origin has to be given.
  3. Fig. 2 should contain error analysis (if available).
  4. Line 170: The equation is not numbered whilst other one (line 229) is.
  5. Fig. 5: Y axis caption has to be rewritten from "Coefficient coeficcient R" to "Correlation coefficient R".
  6. line 245: "R" instead of "®". 
  7. lines 262 and 264: "100×100" instead of "100x100". Special symbol × has to be used here.

All in all, the manuscript is interesting but some corrections should be done in order to improve the quality of the manuscript and consequently to reach the level which is adequate for Coatings journal. 

Author Response

Response to Reviewer 1 Comments

Point 1: The introduction part is too superficial. Introduction does not provide sufficient background and includes only few relevant references. Thus, introduction has to be given in a larger extent and containing additional relevant references.

Response 1: The Introduction already has 9 references. Nevertheless, we have updated the Introduction extensively and added new references, trying to satisfy all reviewers. We also extended the Discussion section, which is now 3 times longer than originally.

Point 2: Lines 108, 109, 130 and 135: For each used instrument, the supplier, city and country of the origin has to be given.

Response 2: We have added the requested information. However, note that Remplir AB does not exist nowadays.

Point 3: Fig. 2 should contain error analysis (if available).

Response 3: The spectrum is just a typical example. However, we added estimated confidence limits in the caption.

Point 4: Line 170: The equation is not numbered whilst other one (line 229) is.

Response 4: The editor missed the number in the process of editing and she said the number of the equation existed in the original edition. It is included in our revised version.

Point 5: Fig. 5: Y axis caption has to be rewritten from "Coefficient coeficcient R" to "Correlation coefficient R".

Response 5: Thank you. The Y axis caption has been rewritten from "Coefficient coeficcient R" to "Correlation coefficient R".

Point 6: line 245: "R" instead of "®".

Response 6: Thank you. The error is typical for the automatic changes in WORD, which we missed to detect here. "®"is corrected to "R". We hope that this error will not occur again when opening the manuscript in another WORD version.

Point 7: lines 262 and 264: "100×100" instead of "100x100". Special symbol × has to be used here.

Response 7: "100x100" is corrected to "100×100".

Reviewer 2 Report

The authors efforts in conducting the research and preparing the manuscript is appreciated. I would have some comments that I hope would help improve the manuscript and make it more suitable for publication: 

  • Introduction can be improved by a brief review of previous art. A paragraph summarizing the highlights of previous efforts to measure and quantify pavement surface friction would be enough. 
  • As the authors know, skid resistance of pavement follows two mechanisms; shear resistance due to molecular adhesion mainly governed by microtexture, and hysteresis due to the mechanical interaction between the tire and macro-texture features. Please clarify how the Mu-meter results and BPN represent these mechanisms. This is important because most readers will use your results in developing computer/mathematical models and need to know how different parameters are related. 
  • Please discuss in the methodology that how the physical measurements (friction) will be related to texture measurements. 
  • Please make some discussions on how your method compares to or aligns with existing standards such ASTM E274 / E274M-15, ASTM E501-08(2015), ASTM E524-08(2015), ASTM E965-15, and ASTM E2380 / E2380M-15. 
  • Discussions section is inadequate. Some of the above-mentioned items can be discussed in this section to improve it, but it is not sufficient. The paper should go beyond a technical report in analyzing and interpreting its results from scientific standpoint. 

Author Response

Response to Reviewer 2 Comments

Point 1: Introduction can be improved by a brief review of previous art. A paragraph summarizing the highlights of previous efforts to measure and quantify pavement surface friction would be enough.

Response 1: We have added text and references, trying to satisfy the comment.

Point 2: As the authors know, skid resistance of pavement follows two mechanisms; shear resistance due to molecular adhesion mainly governed by microtexture, and hysteresis due to the mechanical interaction between the tire and macro-texture features. Please clarify how the Mu-meter results and BPN represent these mechanisms. This is important because most readers will use your results in developing computer/mathematical models and need to know how different parameters are related.

Response 2: We have added three lines to each instrument explaining how they represent these mechanisms. A good idea.

Point 3: Please discuss in the methodology that how the physical measurements (friction) will be related to texture measurements.

Response 3: Well, this is the idea of the study, so it is discussed quite extensively in Chapters 4 and 5. Furthermore, the extra text added in Response 2 will also satisfy Point 3, we think.

Point 4: Please make some discussions on how your method compares to or aligns with existing standards such ASTM E274 / E274M-15, ASTM E501-08(2015), ASTM E524-08(2015), ASTM E965-15, and ASTM E2380 / E2380M-15.

Response 4: Actually, the subject of the study is “Spectral techniques …” and this is not addressed in any of these standards; in fact, in no ASTM standard related to friction. But we have added a paragraph about this at the end of the Discussion.

Point 5: Discussions section is inadequate. Some of the above-mentioned items can be discussed in this section to improve it, but it is not sufficient. The paper should go beyond a technical report in analyzing and interpreting its results from scientific standpoint.

Response 5: The comment is justified. We have therefore extended the Discussion section, which is now 3 times longer than originally.

Reviewer 3 Report

This reviewer has thoroughly reviewed the manuscript entitled “Spectral techniques applied to evaluate pavement friction and surface texture”, written by Yan Y. et al. The study aims to find a correlation between the pavement texture and its friction coefficient. Data were acquired roughly 30 years ago using a stationary laser profilometer, Mu-meter and TRL pendulum tester. Results show that texture spectral levels have a good correlation with friction coefficient, especially the wavelengths at 1.25–12.5 mm threshold versus SFCsl.

The manuscript is well written and could be of great interest for the potential readers of this journal. However, this reviewer has to point some issues that must be addressed by the authors before submitting his final recommendation.

Literature analysis should be expanded. There are few references to previous literature in this field. On this regard, authors can also provide a table with previous methodologies and equations developed by other authors to measure friction coefficient by using MTD, PSD or other indirect parameters, indicating its accuracy.

The authors mention in the text that they used only the dataset “New Zealand”, so they can complete this research by using the 2 remaining datasets for the validation of the proposed model.

Please, add a table or figure with the main properties of the 21 pavement samples considered in the study (e.g.: MTD, MPD, pavement type, mixture type, aggregate type, max. aggregate size, AADT, etc.). A map of the study area allocating them will be of great help as well.

Discussion Section is too short and needs to be extended with a deeper analysis, taking into account the previous suggestions made by this reviewer.

References format should be revised as well.

Finally, this reviewer encourages the authors to publish the detailed measurement datasets as supplementary material, to widen the utility of their research.

Minor remarks:

  • L19: Parameter SFCsl not defined in the abstract.
  • L95: Better use “sand patch” method.
  • L95 & 101: Avoid the use of expressions as “(see below)”. Anyway, refer to a concrete Section, Figure or Table.
  • L170: Please, add an order number to the equation.
  • Fig. 5: Correct the Y-axis label. Define SFCX80…SFCsl or change it for “SFC at 80 km/h” and so on.
  • L245: Susbtitiute ® with (R)

Author Response

Response to Reviewer 3 Comments

Point 1: Literature analysis should be expanded. There are few references to previous literature in this field. On this regard, authors can also provide a table with previous methodologies and equations developed by other authors to measure friction coefficient by using MTD, PSD or other indirect parameters, indicating its accuracy.

Response 1: We have updated the Introduction extensively and added new references. We also extended the Discussion section, as suggested by another reviewer, which is now 3 times longer than originally (including 8 new references).

Point 2: Please, add a table or figure with the main properties of the 21 pavement samples considered in the study (e.g.: MTD, MPD, pavement type, mixture type, aggregate type, max. aggregate size, AADT, etc.). A map of the study area allocating them will be of great help as well.

Response 2: Such tables exist in ref 14, as Tables 1 and 4. It is possible to combine them into one, but it will require two full pages in the article. If this is needed, we will do so. However, considering the difficulties fitting it into the article in a visually pleasing and readable way, we suggest instead to supply these data as an extensive Excel table as a supplement (see Point 5). We ask the editor to decide about how to handle this.

Point 3: Discussion Section is too short and needs to be extended with a deeper analysis, taking into account the previous suggestions made by this reviewer.

Response 3: We have added two new paragraphs in this section.

Point 4: References format should be revised as well.

Response 4: Thank you, We have revised all the references according to the MDPI references guide.

Point 5: Finally, this reviewer encourages the authors to publish the detailed measurement datasets as supplementary material, to widen the utility of their research.

Response 5: Thank you. We will consider this. See Response 2 above.

Point 6: L19: Parameter SFCsl not defined in the abstract.

Response 6: Parameter SFCsl is defined in the abstract now.

Point 7: L95: Better use “sand patch” method.

Response 7: Yes, of course, “sand method” is corrected to“sand patch”method now.

Point 8: L95 & 101: Avoid the use of expressions as “(see below)”. Anyway, refer to a concrete Section, Figure or Table.

Response 8: The expressions“see below”are corrected to “refer to section 4.1”.

Point 9: L170: Please, add an order number to the equation.

Response 9: The editor missed the number in the process of editing and she said the number of the equation existed in the original edition. It is added in this version of the manuscript.

Point 10: Fig. 5: Correct the Y-axis label. Define SFCX80…SFCsl or change it for “SFC at 80 km/h” and so on.

Response 10: The Y axis caption has been rewritten from "Coefficient coeficcient R" to "Correlation coefficient R". SFC40, SFC60, SFC80 are now defined in the figure 5 caption, SFCsl is defined in line 209.

Point 11: L245: Susbtitiute ® with (R).

Response 11:  Thank you. The error is typical for the automatic changes in WORD, which we missed to detect here. "®"is corrected to "R". We hope that this error will not occur again when opening the manuscript in another WORD version.

Reviewer 4 Report

The texture of pavements is important on all roads and should be optimized for optimum car tires traction, low running noise, car stability, low fuel consumption, even in rainy conditions. The subject of the paper is still interesting, even if 30 years have passed since the experimental research and the authors' first report (1990, [13-14]).

In the meanwhile, the techniques of data acquisition advanced, but this research is still valid as the calibration of the old equipment was correctly realized. It seems that, starting from this research, a new project was developed 25 years ago [4].

The paper must be improved to be considered for publication. Consequently, I have some suggestions.

1. The Introduction section cited mostly old papers, reports and conference communications. A thorough literature study must be included in the introduction of the present paper. I indicated some new references at the end of this review. I advise the authors to carefully read them and to update the introduction and all the text of the paper according to the newest published results in the field.

2. Section 2 should present more on spectral analysis theory and the formulas for correlation and intercorrelation functions, explaining their meaning in concordance with the subject of the paper.

3. At line 137, the differences between the old and the new standard have to be emphasized. Also, the authors should comment on the existing new standard indicated by supplementary reference [2]: “DD ISO / TS 13473-4: 2008; Characterization of pavement texture by use of surface profiles. Spectral analysis of texture profiles ”. Is this a new standard based on the findings of the present paper or not?!

4. The results of PIARC study were published in journals till now, or is it just a public report?! At the end of Discussions section, the authors are invited to argue the novelty of their research versus reference [4], the PIARC study.

5. Line 158, wrong reference to Figure 4.

Supplementary references:

1. http://dx.doi.org/10.1080/10298436.2014.972956

2. https://doi.org/10.3403/30182702

3. DOI: 10.3141/2094-15 (https://journals.sagepub.com/doi/abs/10.3141/2094-15 )

4. https://doi.org/10.1016/j.conbuildmat.2015.08.117

5. DOI:10.2478/cee-2014-0015  (https://www.researchgate.net/publication/273303675_The_Usability_of_Different_Skid_Resistance_Characteristics_in_Road_Assessment )

Author Response

Response to Reviewer 4 Comments

Point 1: The Introduction section cited mostly old papers, reports and conference communications. A thorough literature study must be included in the introduction of the present paper. I indicated some new references at the end of this review. I advise the authors to carefully read them and to update the introduction and all the text of the paper according to the newest published results in the field.

Response 1: This paper is not intended to be a literature study. It describes an old study that was unfortunately not published before, but which still provides important information about the influence of texture on friction measurements, in particular spectral properties of the texture profile. If one would cover the most important modern literature on friction and texture, it would be a full article in itself. In our mention of other published literature, we therefore focus on the spectral technology and the MPD which was developed based on this. Otherwise, the spectral technology is not much dealt with in published friction-related literature. However, we recognize and agree on the desire to extend the Introduction and the Discussion and have done so very extensively.

Point 2:  Section 2 should present more on spectral analysis theory and the formulas for correlation and intercorrelation functions, explaining their meaning in concordance with the subject of the paper.

Response 2: These are standard procedures in technical science and are not usually included in scientific articles. They fit better in educational textbooks and other educational literature. Correlation is basic statistics. Also, we have referenced to one IEC standard and one ISO standard regarding spectral analysis. Thus, we think that this article is not the appropriate place to include more about basic data processing.

Point 3: At line 137, the differences between the old and the new standard have to be emphasized. Also, the authors should comment on the existing new standard indicated by supplementary reference [2]: “DD ISO / TS 13473-4: 2008; Characterization of pavement texture by use of surface profiles. Spectral analysis of texture profiles ”. Is this a new standard based on the findings of the present paper or not?!

Response 3: Regarding the first sentence: We do not understand the point. Close to this line we do not compare or mention any “old and new standard”. We assume that this point must refer to another line? If we get this clarified, we will consider the point.

Regarding ISO/TS 13473-4 (which is our ref 3 in the original version), this standard is based on extensive experience by many researchers; of which Dr Sandberg is one (actually he was project leader of developing it), and the study presented here is one small piece of this. But it is misleading to say that the ISO publication is based on this study. We do not suggest it in the text, so we find no need to revise the text. BTW, the ISO/TS 13473-4:2008 is presently subject of an update.

Point 4: The results of PIARC study were published in journals till now, or is it just a public report?! At the end of Discussions section, the authors are invited to argue the novelty of their research versus reference [4], the PIARC study.

Response 4: The PIARC study was reported in an extensive report (book) published by PIARC (400+ pages). Many transportation libraries should have a copy, but as it is difficult to find it digitally, we have added a web link to this report on the PIARC website. The “novelty” of this study is already mentioned in the Discussion as the third paragraph, and now mentioned also in the new last paragraph, which should be enough.

Point 5: Line 158, wrong reference to Figure 4.

Response 5: “Figure 4” is corrected to “Figure 3”.

Supplementary references:

  1. http://dx.doi.org/10.1080/10298436.2014.972956

This is an excellent reference, we included it.

  1. https://doi.org/10.3403/30182702

This is our ref ISO/TS 13473-4:2008 (i.e. nothing new)

  1. DOI: 10.3141/2094-15 (https://journals.sagepub.com/doi/abs/10.3141/2094-15 )

This is about IFI and is now included.

  1. https://doi.org/10.1016/j.conbuildmat.2015.08.117

This is a study with very limited relevance as it measured friction only by BPN. Essentially, it compares ETD and BPN with derived image parameters, so it compares various texture measurement and characterization methods, and not its relation to friction. This has low relevance to our study and is not worth mentioning.

5.DOI:10.2478/cee-2014-0015  (https://www.researchgate.net/publication/273303675_The_Usability_of_Different_Skid_Resistance_Characteristics_in_Road_Assessment )

This article mentions PSD but does not apply it. It tries some other texture parameters, which are of secondary interest in our papers. But the rather limited data set and the use of only the BV11 trailer (results of which are known to be correlated mostly with microtexture and very little with macrotexture), as well as the low R2 found, do not justify mentioning it here.

We thank the reviewer for these constructive comments and references.

Round 2

Reviewer 1 Report

The manuscript is ready to be accepted as it is.

Author Response

Response to Reviewer 1 Comments

Point 1: The introduction part is too superficial. Introduction does not provide sufficient background and includes only few relevant references. Thus, introduction has to be given in a larger extent and containing additional relevant references.

Response 1: The Introduction already has 9 references. Nevertheless, we have updated the Introduction extensively and added new references, trying to satisfy all reviewers. We also extended the Discussion section, which is now 3 times longer than originally.

Point 2: Lines 108, 109, 130 and 135: For each used instrument, the supplier, city and country of the origin has to be given.

Response 2: We have added the requested information. However, note that Remplir AB does not exist nowadays.

Point 3: Fig. 2 should contain error analysis (if available).

Response 3: The spectrum is just a typical example. However, we added estimated confidence limits in the caption.

Point 4: Line 170: The equation is not numbered whilst other one (line 229) is.

Response 4: The editor missed the number in the process of editing and she said the number of the equation existed in the original edition. It is included in our revised version.

Point 5: Fig. 5: Y axis caption has to be rewritten from "Coefficient coeficcient R" to "Correlation coefficient R".

Response 5: Thank you. The Y axis caption has been rewritten from "Coefficient coeficcient R" to "Correlation coefficient R".

Point 6: line 245: "R" instead of "®".

Response 6: Thank you. The error is typical for the automatic changes in WORD, which we missed to detect here. "®"is corrected to "R". We hope that this error will not occur again when opening the manuscript in another WORD version.

Point 7: lines 262 and 264: "100×100" instead of "100x100". Special symbol × has to be used here.

Response 7: "100x100" is corrected to "100×100"

We really appreciate your detailed comments. Thank you very much.

Reviewer 3 Report

The authors have addressed the majority of the comments made by this reviewer. However, some points were not taken into account, and in the opinion of this reviewer they should have been fully considered in the revised manuscript:

Point 1: Authors can also provide a table with previous methodologies and equations developed by other authors to measure friction coefficient by using MTD, PSD or other indirect parameters, indicating its accuracy.

Point 2: Please, add a table or figure with the main properties of the 21 pavement samples considered in the study (e.g.: MTD, MPD, pavement type, mixture type, aggregate type, max. aggregate size, AADT, etc.). If the authors consider that it is too long, they can provide this data as a supplementary material and insert a synthetic table/figure. A map of the study area allocating them will be of great help as well.

Author Response

Response to Reviewer 3 Comments Point 1: Authors can also provide a table with previous methodologies and equations developed by other authors to measure friction coefficient by using MTD, PSD or other indirect parameters, indicating its accuracy. Response 1: Although such a compilation would be interesting, it would need to be very extensive. Accuracies or uncertainties are rarely if ever reported. We would consider that task as almost a state-of-the-art article. We are sorry, but at the moment we think that it would be too much work. We also note that no other reviewer has suggested this. Point 2: Please, add a table or figure with the main properties of the 21 pavement samples considered in the study (e.g.: MTD, MPD, pavement type, mixture type, aggregate type, max. aggregate size, AADT, etc.). If the authors consider that it is too long, they can provide this data as a supplementary material and insert a synthetic table/figure. A map of the study area allocating them will be of great help as well. Response 2: Such tables exist in ref 14, as Tables 1 and 4. It is possible to combine them into one, but it will require two full pages in the article. If this is needed, we will do so. However, considering the difficulties fitting it into the article in a visually pleasing and readable way, we suggest instead to supply these data as an extensive Excel table as a supplement (see Point 5 in the first-round review). The Excel file can be supplemented with the requested map. However, it may be a problem to get a map which is not copyrighted. We ask the editor to decide about how to handle this. If the option with an Excel file is chosen, we would be grateful if it would be possible to get some extra time to produce it (say until 1 June), without further delaying publication of the article.

Reviewer 4 Report

The authors substantially improved their paper, but still keep the article in the form of a research report. 

As the authors asserted, better experimental work was already published in form of a book [4]. I tried to access that book, but I have had no access, because it is restricted to membership of a professional association. This is the reason determining me to recommend this paper for publication in an open access journal.

I expect with interest your further research on the concerned field.

Author Response

Response to Reviewer 4 Comments

The authors substantially improved their paper, but still keep the article in the form of a research report.

As the authors asserted, better experimental work was already published in form of a book [4]. I tried to access that book, but I have had no access, because it is restricted to membership of a professional association. This is the reason determining me to recommend this paper for publication in an open access journal.

I expect with interest your further research on the concerned field.

Authors’ response: Yes, we are aware that this PIARC book may be difficult to access. We of course have both the printed version and a digital version, due to one of the authors taking part in the PIARC study. The lack of open published research of using spectral technology in skid resistance research was the reason why we decided to write an article about it for publication, as it may inspire other researchers to use the spectral technology in this subject.